# mRNA vaccines and hybrid immunity use different B cell germlines against Omicron BA.4 and BA.5

Emanuele Andreano [1], Ida Paciello[1], Giulio Pierleoni[2], Giuseppe Maccari [3], Giada Antonelli[1], Valentina Abbiento [1], Piero Pileri [1], Linda Benincasa[2], Ginevra Giglioli[2], Giulia Piccini [4], Concetta De Santi[1], Claudia Sala[1], Duccio Medini[3], Emanuele Montomoli[2,4,5], Piet Maes [6] & Rino Rappuoli [7,8] ✉

Severe acute respiratory syndrome 2 Omicron BA.4 and BA.5 are characterized by high transmissibility and ability to escape natural and vaccine induced immunity. Here we test the neutralizing activity of 482 human monoclonal antibodies isolated from people who received two or three mRNA vaccine doses or from people vaccinated after infection. The BA.4 and BA.5 variants are neutralized only by approximately 15% of antibodies. Remarkably, the antibodies isolated after three vaccine doses target mainly the receptor binding domain Class 1/2, while antibodies isolated after infection recognize mostly the receptor binding domain Class 3 epitope region and the N-terminal domain. Different B cell germlines are used by the analyzed cohorts. The observation that mRNA vaccination and hybrid immunity elicit a different immunity against the same antigen is intriguing and its understanding may help to design the next generation of therapeutics and vaccines against coronavirus disease 2019.

Almost 3 years after the first case of SARS-CoV-2 in Wuhan in December 2020, 761 million cases and 6.8 million deaths have been reported to be caused by the COVID-19 pandemic. In the meantime, the original Wuhan virus has been replaced by several variants of concern named Alpha, Beta, Gamma, Delta, and Omicron, each characterized by the ability to escape natural and vaccine induced antibody neutralization and by an improved ability to transmit from person to person[1]. Since November 2021 the Omicron variant replaced all previous viruses and generated new lineages named BA.1, BA.2, BA.4, and BA.5[1–3]. According to the global initiative on sharing all influenza data (GISAID) database the most recent SARS-CoV-2 Omicron sublineages BA.4 and BA.5 were the most abundant SARS-CoV-2 circulating variants worldwide until october-november 2022[4]. Different scenarios could explain the fast spread of these new sublineages and the ability to outcompete previous Omicron variants. Examples are the lack of the G496S mutation in the spike (S) protein, which results in increased human angiotensin-converting enzyme 2 binding affinities compared to other Omicron variants[5], the evolution of novel mutations on the S protein, which conferred enhanced resistance to neutralizing antibodies[6], and the ability to better suppress and antagonize the innate immune defenses[7]. SARS-CoV-2 neutralizing antibodies target the receptor binding domain (RBD) and N terminal domain (NTD) of the S protein, which is a trimeric glycoprotein exposed on the surface of the virus[8,9]. The SARS-CoV-2 Omicron sublineages BA.4 and BA.5 share an identical S glycoprotein which carries 31 mutations on its surface (Supplementary Fig. 1a)[6]. As for the initial Omicron BA.1 and BA.2, both the RBD and

[1]Monoclonal Antibody Discovery (MAD) Lab, Fondazione Toscana Life Sciences, Siena, Italy. [2]VisMederi Research S.r.l., Siena, Italy. [3]Data Science for Health (DaScH) Lab, Fondazione Toscana Life Sciences, Siena, Italy. [4]VisMederi S.r.l, Siena, Italy. [5]Department of Molecular and Developmental Medicine, University of Siena, Siena, Italy. [6]KU Leuven, Rega Institute, Department of Microbiology, Immunology and Transplantation, Laboratory of Clinical and Epidemiological Virology, Leuven, Belgium. [7]Department of Biotechnology, Chemistry and Pharmacy, University of Siena, Siena, Italy. [8]Fondazione Biotecnopolo di Siena, Siena, Italy. ✉e-mail: rino.rappuoli@biotecnopolo.it

NTD immunodominant sites of these new sublineages are heavily mutated. The NTD of BA.4 and BA.5 harbors 9 mutations (29.0%) on this domain which are represented by 4 substitutions (T19I, L24S, G142D, and V213G) and 5 deletions (Δ25−27 and Δ69−70). The mutational pattern is extremely similar to the parental BA.2 lineage with the exception of the Δ69-70 mutation which was present in the original BA.1 Omicron variant. As observed in all previous SARS-CoV-2 variants of concern, the RBD of Omicron BA.4 and BA.5 displays only substituted residues highlighting the more conservative structure of this domain. The RBD carries 17 mutations (54.8%), and 9 of them are within the receptor binding motif (RBM) which spans from residue S438 to Y508 of the S protein[10] (Supplementary Fig. 1b). Recent studies have described the impact of the Omicron variants, including BA.4 and BA.5, on the polyclonal antibody response of subjects infected, vaccinated and with hybrid immunity[11–13], as well as on a set of 28 and 158 neutralizing antibodies (including therapeutic, database and previous publication-derived antibodies isolated from a variety of subjects and cohorts)[14,15], or on a library of 1640 RBD-binding antibodies[5]. In this study we evaluated the neutralizing activity against BA.4 and BA.5 variants of 482 neutralizing human monoclonal antibodies (nAbs) that neutralized the original Wuhan virus. Our data confirm at single cell level that only a minority of nAbs cross-neutralize BA.4 and BA.5 lineages and reveal that the B cell germlines usage and S protein epitopes targeted for cross-neutralization are different in vaccinated and infected people.

## Results

### Neutralization of Omicron sublineages

A collection of 482 nAbs against the SARS-CoV-2 virus originally isolated in Wuhan, were used in this study. They derived from three different cohorts: SARS-CoV-2 seronegative subjects vaccinated with two

(SN2; $n = 5$) or three (SN3; $n = 4$) doses of COVID-19 mRNA vaccines, and subjects exposed to SARS-CoV-2 infection and subsequently vaccinated with two doses of the same mRNA vaccines (seropositive 2nd dose; SP2; $n = 5$)[16,17]. The four subjects in the SN3 cohort (VAC-001, VAC-002, VAC-008, and VAC-010) are the same analyzed in the SN2 group. All subjects were enrolled in our previous studies and gave their written consent. In addition, all subjects, with the exception of VAC-010 in the SN3 cohort which was immunized with mRNA-1273, received the BNT162b2 mRNA vaccine[16,17]. Their neutralizing potency against the SARS-CoV-2 Omicron variants BA.4 and BA.5, was tested by the cytopathic effect-based microneutralization assay (CPE-MN) against live viruses in biosafety level 3 (BSL3) laboratories. Overall, less than 15% of the antibodies retained neutralizing activity against the Omicron BA.4 and BA.5 variants. As shown in Fig. 1, none of the 52 antibodies from the SN2 cohort were able to neutralize Omicron BA.4 and BA.5 variants, while minimal cross-protection was observed against BA.1 ($n = 1$; 1.9%) and BA.2 ($n = 4$; 7.7%) (Fig. 1a). Conversely, of the 206 nAbs in the SN3 cohort, 14.6 ($n = 30$) and 14.1% ($n = 29$) cross-neutralized Omicron BA.4 and BA.5, respectively (Fig. 1b). No major differences in numbers and frequency of nAbs in the SN3 cohort were noticed between the three vaccinees immunized with the BNT162b vaccine and the subjects which received the mRNA-1273 booster dose (Supplementary Table 1). Similarly, in the case of SP2, 15.5 ($n = 32$) and 14.6% ($n = 30$) of the 224 nAbs cross-neutralized these SARS-CoV-2 Omicron variants (Fig. 1c). The overall nAbs neutralization potency against all tested Omicron variants was also evaluated and reported as geometric mean 100% inhibitory concentration (GM-IC100). Not-neutralizing antibodies were excluded from the GM-IC100 analysis. Compared to the neutralization GM-IC100 observed against the Wuhan virus, we observed a 1.62-, 1.66-, 2.62-, and 2.37-fold decrease against the BA.1, BA.2, BA.4 and BA.5 respectively in the SN3 cohort, and a 3.16-,

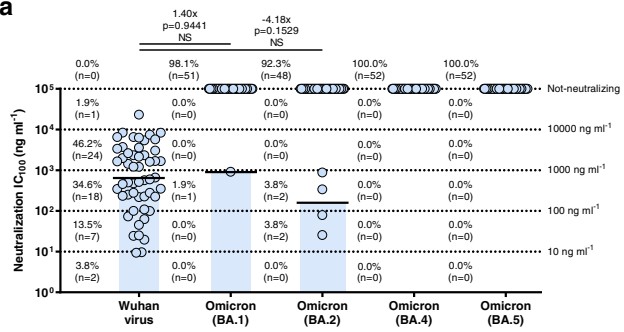

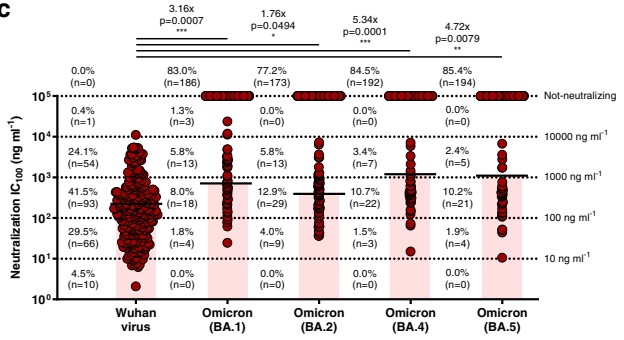

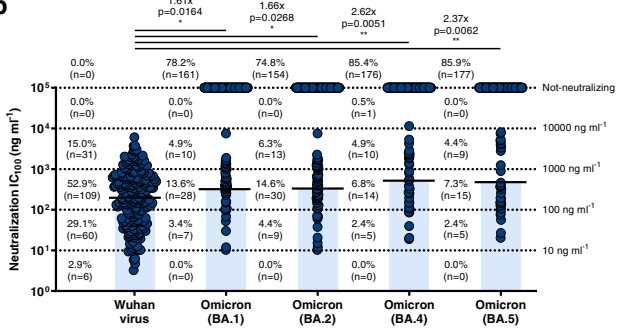

**d**

| Cohort | Wuhan virus | Omicron (BA.1) | Omicron (BA.2) | Omicron (BA.4) | Omicron (BA.5) |
|---|---|---|---|---|---|
| Seronegative 2nd dose (SN2) GM-IC100 ng ml⁻¹ ($n = 52$) | 654.5 | 919.9 | 156.7 | N/A | N/A |
| Seronegative 3rd dose (SN3) GM-IC100 ng ml⁻¹ ($n = 206$) | 201.4 | 324.9 | 334.4 | 528.0 | 476.4 |
| Seropositive 2nd dose (SP2) GM-IC100 ng ml⁻¹ ($n = 224$) | 227.5 | 719.8 | 401.5 | 1215.3 | 1074.1 |

**Fig. 1 | Potency and breadth of neutralization of nAbs against SARS-CoV-2 Omicron variants.** Scatter dot charts show the neutralization potency, reported as IC100 (ng ml⁻¹), of nAbs tested against the original Wuhan SARS-CoV-2 virus, and the Omicron BA.1, BA.2, BA.4 and BA.5 lineages for SN2 (**a**), SN3 (**b**), and SP2 (**c**), respectively. The number, percentage, GM-IC100 (black lines and colored bars), fold-change and statistical significance of nAbs are denoted on each graph. Reported fold-change and statistical significance are in comparison with the Wuhan virus. Technical duplicates were performed for each experiment. **d** The table shows the IC100 geometric mean (GM-IC100) of all nAbs pulled together from SN2, SN3 and SP2 against all SARS-CoV-2 viruses tested. A nonparametric Mann–Whitney $t$ test was used to evaluate statistical significances between groups. Two-tailed $p$ value significances are shown as *$p < 0.05$, **$p < 0.01$, and ***$p < 0.001$. Source data are provided as a Source Data file.

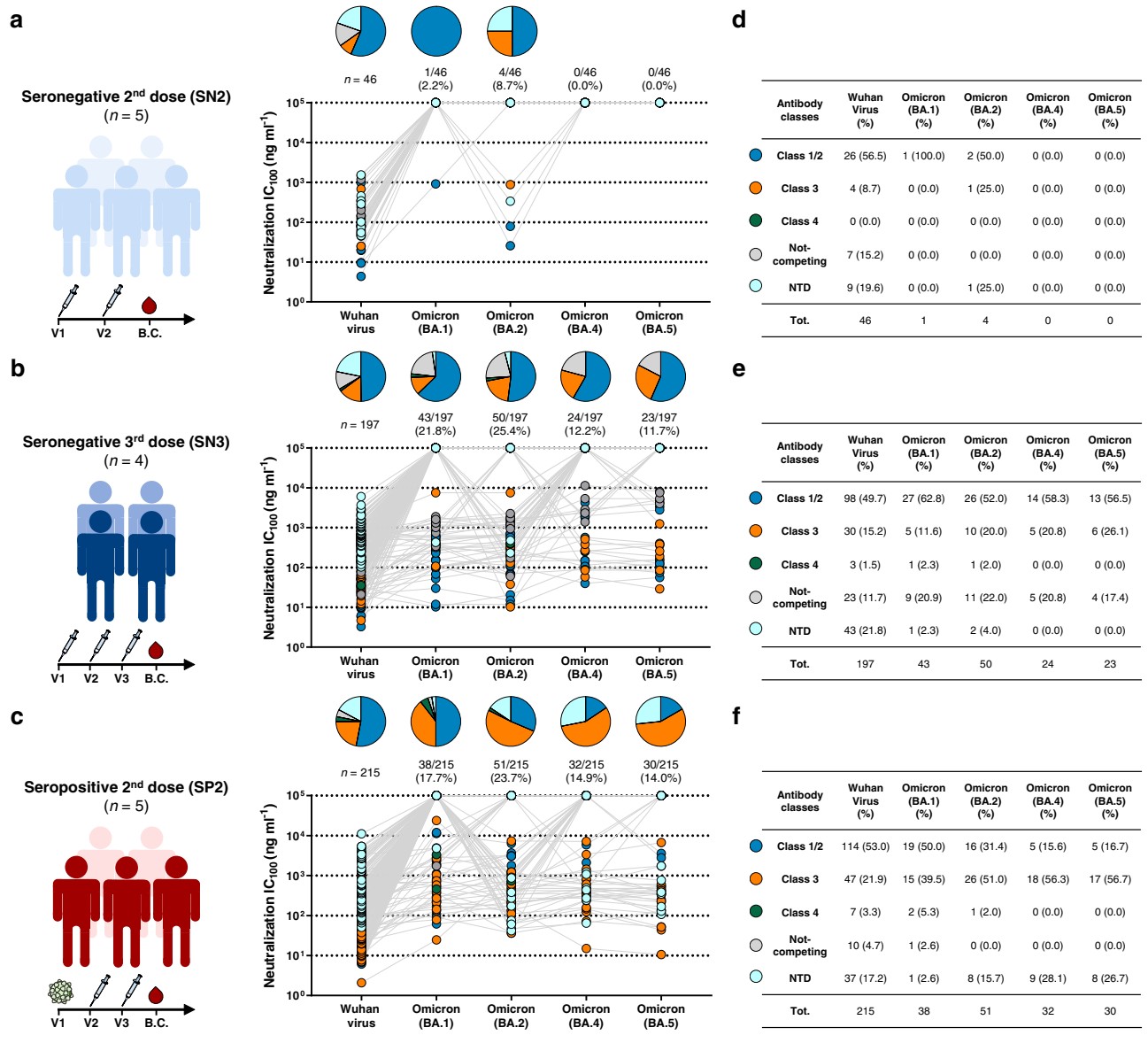

**Fig. 2 | Distribution of RBD and NTD-targeting nAbs against Omicron variants.** Pie charts show the distribution of cross-protective nAbs based on their ability to bind Class 1/2 (blue), Class 3 (orange) and Class 4 (dark green) regions on the RBD, as well as not-competing nAbs (gray) and NTD-targeting nAbs (cyan). Dot charts show the neutralization potency, reported as IC$_{100}$ (ng ml$^{-1}$), of nAbs against the Wuhan virus and the Omicron BA.1, BA.2, BA.4 and BA.5 variants observed in the SN2 (**a**), SN3 (**b**) and SP2 (**c**) cohorts. The number and percentage of nAbs are denoted on each graph. Tables summarize number and percentage of Class 1/2, Class 3, Class 4, not-competing and NTD-targeting nAbs for each tested variant in the SN2 (**d**), SN3 (**e**) and SP2 (**f**) cohorts. Source data are provided as a Source Data file.

1.76-, 5.34-, and 4.72-fold decrease against the BA.1, BA.2, BA.4 and BA.5, respectively in the SP2 group (Fig. 1b–d). Interestingly, none of the nAbs tested showed extremely potent neutralization activity (IC$_{100}$ < 10 ng ml$^{-1}$) against all Omicron viruses.

## Mapping RBD and NTD cross-protective nAbs

To understand the type of antibodies mainly responsible for cross-protection against the Omicron variants, we investigated the neutralization activity of RBD and NTD binding nAbs (Fig. 2). Antibodies which did not bind to the RBD or NTD (6, 9, and 9 nAbs for SN2, SN3 and SP2 respectively) were not included in this analysis. RBD-targeting nAbs for all cohorts were previously classified based on their ability to compete with the Class 1/2 antibody J08[18], the Class 3 antibody S309[19], and the Class 4 antibody CR3022[20], or for their lack of competition with the three tested antibodies (Not-competing)[17,21]. We previously observed that each individual shows heterogenicity within their respective antibody response, but similar trends of classes distribution were observed among subjects within the same cohort[16,17]. The SN2 group (n = 46) showed mainly nAbs targeting the RBD-Class 1/2 epitope region against Wuhan (n = 26; 56.5%), BA.1 (n = 1; 100%) and BA.2 (n = 2; 50%), while no neutralization activity was observed against the BA.4 and BA.5 Omicron variants (Fig. 2a,d). As expected, a similar trend was also observed for RBD and NTD-targeting nAbs isolated from the SN3 cohort (n = 197). Indeed, RBD-Class 1/2 targeting nAbs represented the 49.7% (n = 98) of antibodies neutralizing the Wuhan virus, and their percentage increased against the Omicron sublineages constituting the 62.8 (n = 27), 52.0 (n = 26), 58.3 (n = 14) and 56.5% (n = 13) of nAbs able to cross-neutralize the Omicron BA.1, BA.2, BA.4 and BA.5 respectively (Fig. 2b,e). Interestingly, while NTD-targeting antibodies were the second most abundant class among Wuhan nAbs, they lost almost completely their functionality against the Omicron lineages, representing only 2.3 (n = 1) and 4.0% (n = 2) of nAbs against BA.1 and

BA.2, while showing no activity towards the Omicron BA.4 and BA.5 (Fig. 2b,e). Interestingly, RBD and NTD-targeting nAbs isolated from the SP2 cohort ($n = 215$) showed a completely different profile against the Omicron variants. In fact, cross-neutralizing antibodies targeted preferentially the RBD-Class 3 epitope region and constituted the 51.0 ($n = 26$), 56.3 ($n = 18$) and 56.7% ($n = 17$) of nAbs able to neutralize Omicron BA.2, BA.4 and BA.5 respectively (Fig. 2c,f). In addition, differently from what observed in the SN3 cohort, NTD-targeting nAbs isolated in the SP2 group retained high level of functionality against the Omicron BA.2, BA.4 and BA.5, representing the 15.7 ($n = 8$), 28.1 ($n = 9$) and 26.7% ($n = 8$) cross-protective nAbs against these variants respectively (Fig. 2c,f). The RBD-Class 1/2 antibodies that were the most abundant in neutralizing Wuhan ($n = 114$; 53.0%) and BA.1 ($n = 19$; 50.0%), were heavily escaped by Omicron BA.2, BA.4 and BA.5, representing only the 31.4 ($n = 16$), 15.6 ($n = 5$) and 16.7% ($n = 5$) of nAbs respectively (Fig. 2c,f). Finally, independently from their overall frequency, the neutralization potency of Omicron BA.4 and BA.5 of Class 1/2 and Class 3 nAbs in the SN3 group was higher than in the SP2 group (Supplementary Fig. 2). Noteworthy, while no NTD-targeting antibodies able to neutralize BA.4 and BA.5 were found in the SN3 group, nAbs isolated from the SP2 cohort that targeted this S protein domain were the second most abundant group of antibodies and showed a neutralization potency comparable to Class 3 nAbs and up to 2.44-fold higher GM-IC$_{100}$ compared to Class 1/2 antibodies isolated in this cohort (Supplementary Fig. 2).

## B cell germline usage of cross-neutralizing Omicron nAbs

In addition to the functional characterization and epitope mapping analyses, we investigated the B cell germline and V-J gene rearrangements (IGHV;IGHJ) used by highly cross-reactive nAbs against Omicron variants. Of the 482 nAbs assessed in this study we previously recovered 430 heavy chain sequences: 46 from SN2, 176 from SN3, and 208 from SP2[16,17]. In SN2 subjects, predominant B cell germlines neutralizing the Wuhan strain include IGHV1-69;IGHJ4-1, IGHV3-30;IGHJ6-1, IGHV3-53;IGHJ6-1, IGHV3-66;IGHJ4-1 B cell germlines[16,22–25]. These germlines constituted the 28.3% of nAbs able to neutralize the Wuhan virus and all of them lost completely their functional activity against all Omicron variants; the only exception was one nAb encoded by the IGHV3-53;IGHJ6-1 germline which was able to neutralize with high potency BA.2 (Fig. 3a; Supplementary Table 2; Supplementary Table 3). Differently from the SN2 group, the SN3 and SP2 cohorts contained SARS-CoV-2 cross-neutralizing nAbs against all Omicron variants. In the SN3 cohort, Omicron cross-neutralizing antibodies were dominated by five V-J gene rearrangements. These were IGHV1-58;IGHJ3-1, IGHV1-69;IGHJ3-1, IGHV1-69;IGHJ4-1, IGHV3-66;IGHJ4-1, and IGHV3-66;IGHJ6-1 (Fig. 3b; Supplementary Table 2). These five germlines are well known and encode for potently neutralizing RBD-targeting Class 1 and Class 2 nAbs[20,21,23,26]. These germlines represented the 32.4% of nAbs against the original Wuhan virus, and the 58.3, 54.8, 54.6, and 54.6% of cross-neutralizing nAbs against Omicron BA.1, BA.2, BA.4, and BA.5, respectively (Fig. 3b; Supplementary Table 2). When we analyzed the distribution of these germlines we observed that the IGHV1-69;IGHJ4-1 was the most abundant against all Omicron variants, while nAbs encoded by the IGHV3-66;IGHJ6-1 V-J genes were the only to maintain a GM-IC$_{100}$ against all Omicron variants similar to what observed for the original Wuhan virus (Fig. 3b; Supplementary Table 3). The remaining germlines showed a 1.67- to 45.43-fold reduction in their GM-IC$_{100}$ against the Omicron variants tested in this study compared to the Wuhan virus. For the SP2 cohort, Omicron cross-functional antibodies derived mainly from three germlines which differed from those found in the SN3 group. These germlines used the IGHV1-24;IGHJ6-1, IGHV1-58;IGHJ3-1 and IGHV2-5;IGHJ4-1 V-J gene rearrangements. nAbs encoded by these B cell germlines represent only 11.5% of all antibodies against the Wuhan virus (Fig. 3c; Supplementary Table 2) and their frequency increased to 24.3, 28.6,

30.0 and 31.0% for cross-neutralizing nAbs against Omicron BA.1, BA.2, BA.4 and BA.5 respectively. The IGHV1-58;IGHJ3-1 and IGHV2-5;IGHJ4-1 germlines encoded for RBD-targeting Class 1 and Class 3 nAbs respectively[16,21,26], while the IGHV1-24;IGHJ6-1 V-J gene rearrangement is mainly used by NTD-targeting antibodies[27,28]. With the exception of BA.1, the IGHV2-5;IGHJ4-1 germline is the most frequently used by Omicron cross-neutralizing nAbs isolated in this cohort. In addition, antibodies carrying the IGHV2-5;IGHJ4-1 rearrangement showed to be the only group of nAbs, among the three highly frequent germlines in the SP2 cohort, able to cross-neutralize all Omicron variants although showing a 7.04–13.51-fold decrease in GM-IC$_{100}$ compared to the Wuhan virus (Supplementary Table 2; Supplementary Table 3).

## Impact on therapeutic nAbs

Since the RBD and RBM are heavily mutated in the Omicron BA.4 and BA.5 variants, and they represent the major targets of antibodies approved for clinical treatment of COVID-19, we evaluated the impact of Omicron BA.4 and BA.5 mutations on eight therapeutic mAbs approved for therapy. Specifically, we tested three Class 1 mAbs, REGN10933 (Casirivimab)[29], LY-CoV016 (Etesevimab)[30], and COV2-2196 (Tixagevimab)[31], the Class 2 targeting nAb LY-CoV555 (Bamlanivimab)[32], and four Class 3 directed nAbs, S309 (Sotrovimab)[19], REGN10987 (Imdevimab)[29], LY-CoV1404 (Bebtelovimab)[33] and COV2-2130 (Cilgavimab)[31], by CPE-MN against the live SARS-CoV-2 virus originated in Wuhan and the Omicron BA.1, BA.2, BA.4, and BA.5 variants (Fig. 4). All tested nAbs showed neutralization activity against the ancestral Wuhan virus with a 100% inhibitory concentration (IC$_{100}$) ranging from 19.5 to 176.8 ng ml$^{-1}$. Class 1 and Class 2 nAbs, derived from the IGHV3-11;IGHJ4-1 (Casirivimab), IGHV3-66;IGHJ4-1 (Etesevimab), IGHV1-58;IGHJ3-1 (Tixagevimab), and IGHV1-69;IGHJ6-1 (Bamlanivimab) germlines, were evaded by all Omicron variants. Differently, Class 3 antibodies, encoded by the IGHV1-18;IGHJ4-1 (Sotrovimab), IGHV3-30;IGHJ4-1 (Imdevimab), IGHV2-5;IGHJ1-1 (Bebtelovimab), and IGHV3-15;IGHJ4-1 (Cligavimab) germlines, retained their neutralization activity against at least one Omicron variant. Indeed, S309 (Sotrovimab) was able to neutralize the Omicron BA.1 virus with a 3.17-fold reduction, while no activity was detected against the other Omicron variants. REGN10987 (Imdevimab) and COV2-2130 (Cilgavimab), neutralized three out of four variants despite showing up to 81.31- and 5.65-fold decrease in their respective IC$_{100}$. Finally, LY-CoV1404 (Bebtelovimab), was the only antibody with high neutralization potency against all Omicron lineages showing an IC$_{100}$ of 11.1, 15.6, 44.2, and 62.5 ng ml$^{-1}$ against Omicron BA.1, BA.2, BA.4, and BA.5, respectively (Fig. 4). These results are in line with previously published works[5,14,15,33].

## Discussion

In this work, we took advantage of our unique panel of 482 SARS-CoV-2 neutralizing human monoclonal antibodies to address at single cell level the cross-neutralizing properties against the Omicron variants of B cells induced by vaccine or hybrid immunity. Our nAb panel, built during the last 3 years of the COVID-19 pandemic, was identified from people receiving two or three mRNA vaccine doses, and from SARS-CoV-2 infected people that had been subsequently vaccinated with the BNT162b2 mRNA vaccine[16,17]. In agreement with previous studies performed mainly with whole sera[5,12–14], we observed that two mRNA vaccine doses were not sufficient to mount a protective antibody response against Omicron variants. Conversely, three mRNA vaccine doses and hybrid immunity showed to induce similar, although limited, levels of protection against the Omicron variants, with an overall average of 18.9 and 17.5% of nAbs still able to neutralize these viruses for SN3 and SP2, respectively. The observation that vaccination and hybrid immunity show similar level of Omicron cross-protection is not aligned with previous studies which, through the analyses of the polyclonal response of subjects with heterologous history of vaccination and infection or by PCR analyses to evaluate the effectiveness of

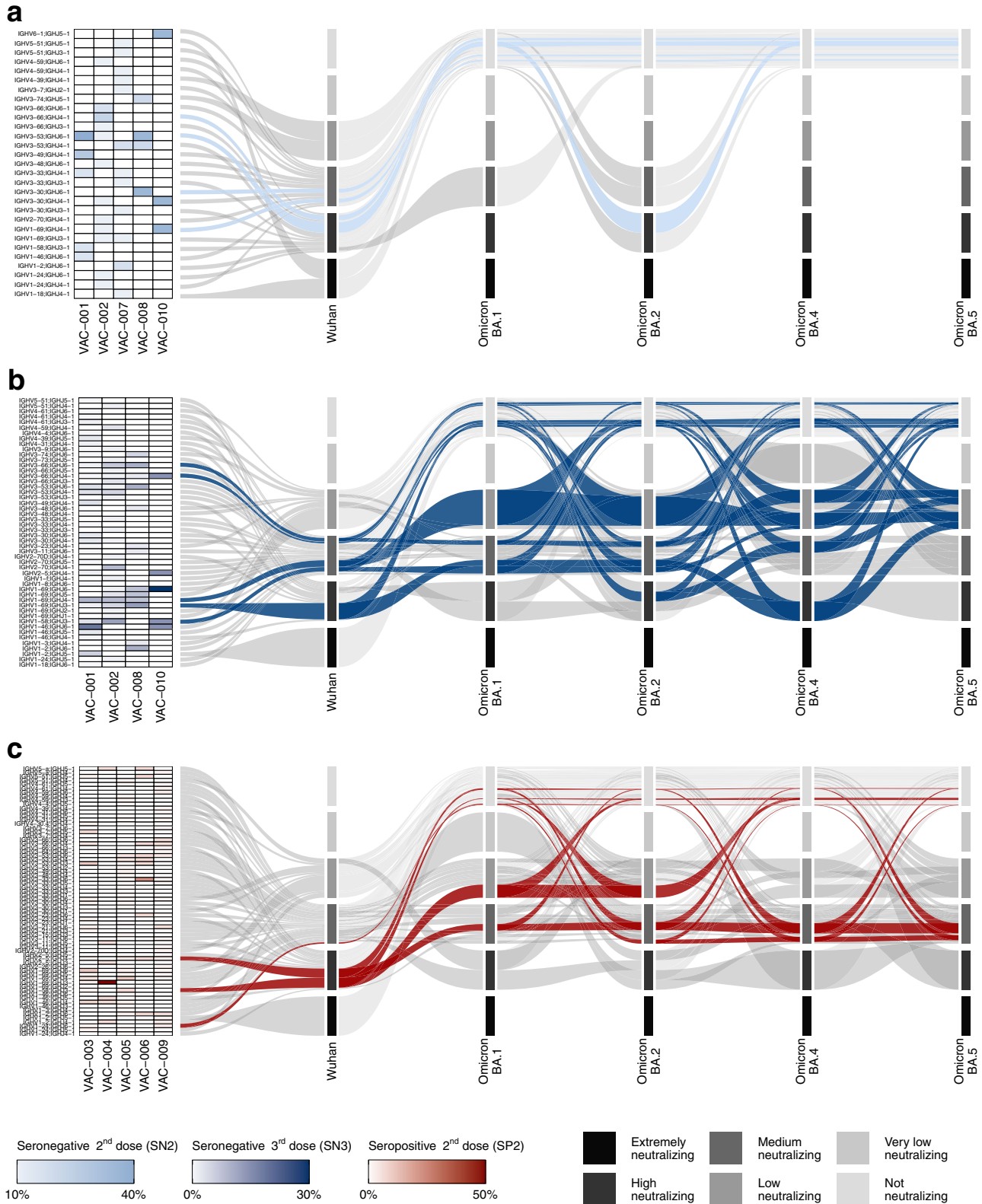

previous infection against reinfection with BA.4 or BA.5, reported higher protection in this latter cohort[5,14,34]. This difference may be given by the different approaches used to evaluate the antibody response of these groups. Indeed, in our study we interrogated exclusively the repertoire of neutralizing antibodies while previous works evaluated the polyclonal antibody response or percentage of reinfection in subjects vaccinated or with hybrid immunity. Another possible explanation for the discrepancy with previous studies is the

low number of participants herein characterized which is the limit of this work. Despite the similarity in the number of antibodies neutralizing the Omicron variants BA.4 and BA.5 in SN3 and SP2, our analyses revealed dramatic differences in the antibody and B cell germline profiles behind their respective responses. Three mRNA vaccine doses expanded mainly RBD-targeting Class 1/2 nAbs and showed a more clonal B cell response constituted mainly by five germlines (IGHV1-58;IGHJ3-1, IGHV1-69;IGHJ3−1, IGHV1-69;IGHJ4-1,

**Fig. 3 | IGHV;IGHJ gene usage of Omicron cross-neutralizing antibodies.** Heatmaps and alluvial plots display the antibody IGHV;IGHJ gene rearrangements frequency for each single donors and for pulled nAbs respectively for SN2 (**a**), SN3 (**b**) and SP2 (**c**). The heatmap on the left represents germline frequency for each individual subject. In the alluvial plots, V-J gene rearrangements were highlighted if they represented at least 10% of all antibodies able to cross-neutralize a specific Omicron variant. Below this threshold, several germlines showed identical or similar frequency values and therefore were not considered as predominant. Selected germlines were highlighted as light blue (IGHV1-69;IGHJ4-1, IGHV3-30;IGHJ6-1, IGHV3-53;IGHJ6-1, IGHV3-66;IGHJ4-1), dark blue (IGHV1-58;IGHJ3-1, IGHV1-69;IGHJ3-1, IGHV1-69;IGHJ4-1, IGHV3-66;IGHJ4-1, and IGHV3-66;IGHJ6-1) and red (IGHV1-24;IGHJ6-1, IGHV1-58;IGHJ3-1, and IGHV2-5;IGHJ4-1) for SN2, SN3, and SP2 respectively. Germline usage is shown for nAbs against the original Wuhan virus and Omicron BA.1, BA.2, BA.4, and BA.5 variants. For each variant, nAbs were grouped into six different categories (strata) based on the neutralization potency (GM-IC$_{100}$) of all nAbs encoded by the specific germline. Strata are defined as extremely neutralizing (≤10 ng ml$^{-1}$), high neutralizing (≤100 ng ml$^{-1}$), medium neutralizing (≤1000 ng ml$^{-1}$), low neutralizing (≤10,000 ng ml$^{-1}$), very low neutralizing (<100,000 ng ml$^{-1}$) and not neutralizing (≥100,000 ng ml$^{-1}$). The flow size indicates the frequency of the specific germline within the strata to which it is linked. Source data are provided as a Source Data file.

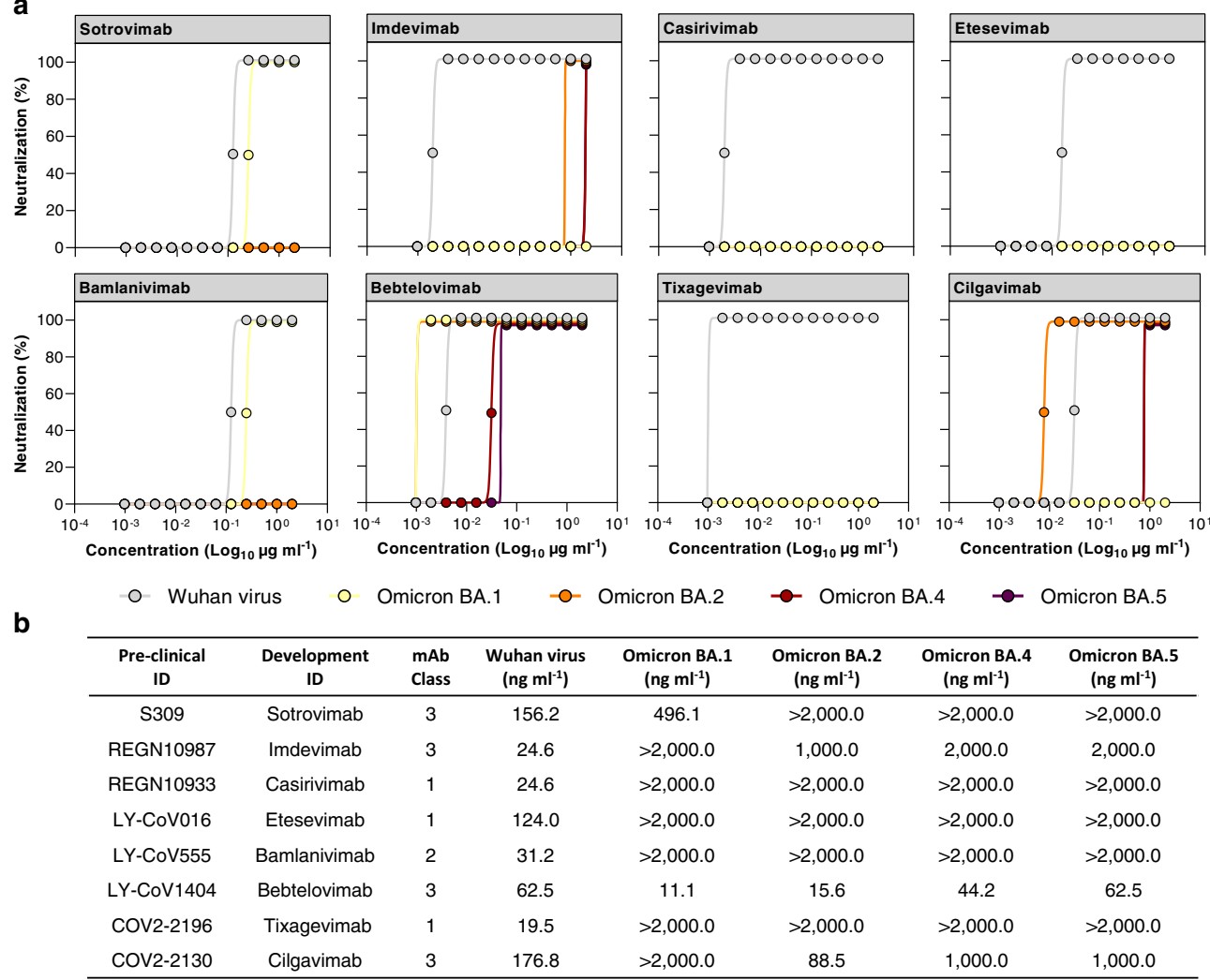

**Fig. 4 | Neutralization activity of COVID-19 therapeutic nAbs. a** Graphs show the CPE-MN neutralization activity of therapeutic nAbs against the original SARS-CoV-2 virus originated in Wuhan and the Omicron BA.1, BA.2, BA.4 and BA.5 variants.

| Pre-clinical ID | Development ID | mAb Class | Wuhan virus (ng ml$^{-1}$) | Omicron BA.1 (ng ml$^{-1}$) | Omicron BA.2 (ng ml$^{-1}$) | Omicron BA.4 (ng ml$^{-1}$) | Omicron BA.5 (ng ml$^{-1}$) |
|---|---|---|---|---|---|---|---|
| S309 | Sotrovimab | 3 | 156.2 | 496.1 | >2,000.0 | >2,000.0 | >2,000.0 |
| REGN10987 | Imdevimab | 3 | 24.6 | >2,000.0 | 1,000.0 | 2,000.0 | 2,000.0 |
| REGN10933 | Casirivimab | 1 | 24.6 | >2,000.0 | >2,000.0 | >2,000.0 | >2,000.0 |
| LY-CoV016 | Etesevimab | 1 | 124.0 | >2,000.0 | >2,000.0 | >2,000.0 | >2,000.0 |
| LY-CoV555 | Bamlanivimab | 2 | 31.2 | >2,000.0 | >2,000.0 | >2,000.0 | >2,000.0 |
| LY-CoV1404 | Bebtelovimab | 3 | 62.5 | 11.1 | 15.6 | 44.2 | 62.5 |
| COV2-2196 | Tixagevimab | 1 | 19.5 | >2,000.0 | >2,000.0 | >2,000.0 | >2,000.0 |
| COV2-2130 | Cilgavimab | 3 | 176.8 | >2,000.0 | 88.5 | 1,000.0 | 1,000.0 |

Technical triplicates were performed for each experiment. **b** The table summarizes the neutralization potency of tested nAbs reported as IC$_{100}$ ng ml$^{-1}$. Source data are provided as a Source Data file.

IGHV3-66;IGHJ4-1, and IGHV3-66;IGHJ6-1) which represented almost 60% of all Omicron cross-neutralizing antibodies. Interestingly, two of the five abundant germlines encoding for cross-neutralizing nAbs in SN3 (IGHV1-69;IGHJ4-1, IGHV3-66;IGHJ4-1) were also present in the SN2 cohort where they showed no functional activity against Omicron variants. This suggests that a third mRNA vaccine dose enhances B cell affinity maturation of selected germlines and drives their expansion and subsequent production of cross-protective nAbs. As for hybrid immunity, RBD-directed Class 3 nAbs and NTD-targeting nAbs were preferentially used and a more diversified B cell response was observed. In fact, only three major germlines were identified (IGHV1-24;IGHJ6-1, IGHV1-58;IGHJ3-1, and IGHV2-5;IGHJ4-1) which represented no more than 31% of the whole antibody response against the Omicron variants. The observation that homologous mRNA vaccination and infection drive the expansion of different B cell germlines which produce nAbs targeting distinct epitopes on the SARS-CoV-2 S protein raises interesting questions about the mechanistic of antigen presentation. Indeed, in both cases the antigen is produced by the host cells and differences in its presentation to the immune cells is likely to derive from the cell types expressing the antigen, the stabilization of

the S protein in its prefusion conformation following the insertion of two prolines, the absence of other viral components and the inflammatory environment present during infection. In addition to the single cell level analysis of our nAb panel, in our study we evaluated the impact of Omicron variants on therapeutic mAbs approved for the treatment of COVID-19 and showed alarming antibody evasion properties by BA.4 and BA.5 against which only one nAb (LY-CoV1404; bebtelovimab) retained high neutralization activity. These results highlight the need to discover novel and broadly reactive monoclonal antibodies able to recognize conserved regions on the S protein which are shared across SARS-CoV-2 variants and among other coronaviruses. Furthermore, the identification of highly conserved regions on the S protein can lead to the structural-based design of new antigens able to elicit a broadly reactive antibody response efficacious against current and future SARS-CoV-2 variants. This strategy could also help to overcome the limits of updating current vaccines with the S protein of emerging variants, like the bivalent BA.4/5 vaccine, which so far did not show to induce superior neutralization titers and breadth against Omicron variants when given as booster compared to a fourth dose of the original vaccine based on the S protein of the Wuhan virus[35,36]. Overall, our work provides unique information on the B cell and antibody response induced by vaccination and infection, highlighting similarities and key differences between these two immunologically distinct cohorts that could be exploited for the design of next generation therapeutics and vaccines against SARS-CoV-2.

## Methods

### Enrollment of COVID-19 vaccinees and human sample collection

Human samples from SARS-CoV-2 infected and vaccinated donors, who received two or three vaccine doses, of both sexes, were previously collected through a collaboration with the Azienda Ospedaliera Universitaria Senese, Siena (IT)[16,17]. All subjects enrolled gave their written consent. The study that allowed the enrollment of subjects in all three cohorts was approved by the Comitato Etico di Area Vasta Sud Est (CEAVSE) ethics committees (Parere 17065 in Siena) and conducted according to good clinical practice in accordance with the declaration of Helsinki (European Council 2001, US Code of Federal Regulations, ICH 1997). This study was unblinded and not randomized. Five subjects were enrolled in both the seronegative 2nd dose (SN2; $n = 5$) and seropositive 2nd dose (SP2; $n = 5$) cohorts[16]. Subjects in the SP2 cohort resulted positive to SARS-CoV-2 infection between October and November 2020 when in Italy only the D614G SARS-CoV-2 variant was circulating[4,16]. The severity of infection of donors in the SP2 cohort ranged from asymptomatic (VAC-003 and VAC-006), to mild (VAC-004 and VAC-005) to moderate (VAC-009). Long-COVID was not reported for any of the donors enrolled in the SP2 cohort. All subjects in the SN2 and SP2 cohorts received two doses of the BNT162b2 mRNA vaccine[16]. Four subjects were enrolled in the seronegative 3rd dose cohort (SN3; $n = 4$), three of which were boosted with the BNT162b2 mRNA vaccine (VAC-001, VAC-002, and VAC-008) and one (VAC-010) received the mRNA-1273 mRNA vaccine[17]. The four seronegative donors in the SN3 cohort also participated in our previous study after two doses of BNT162b2 mRNA vaccine (SN2). The female to male ratio for each cohort was 2:3, 2:2, and 2:3 for SN2, SN3, and SP2, respectively. The age of donors enrolled in the three cohorts ranged from 25 to 57 years (GM of 41.3, 43.2, and 36.2 years of age for SN2, SN3, and SP2, respectively)[16,17]. No statistical methods were used to predetermine sample size.

### SARS-CoV-2 live viruses neutralization assay

All SARS-CoV-2 live virus neutralization assays were performed in the BSL3 laboratories at Toscana Life Sciences in Siena (Italy) and Vismederi Srl, Siena (Italy), which are approved by a Certified Biosafety Professional and inspected annually by local authorities. To assess the neutralization potency and breadth of nAbs against the live SARS-CoV-2 viruses, a cytopathic effect-based microneutralization assay (CPE-MN) was performed as previously described[16,17,21,22]. Briefly, nAbs were co-incubated with SARS-CoV-2 viruses used at 100 median tissue culture infectious dose (100 TCID$_{50}$) for 1 h at 37 °C, 5% CO$_2$. The mixture was then added to the wells of a 96-well plate containing a subconfluent Vero E6 cell monolayer. Plates were incubated for 3–4 days at 37 °C in a humidified environment with 5% CO$_2$, then examined for CPE by means of an inverted optical microscope by two independent operators. TAP expressed nAbs were tested at a starting dilution of 1:5 and diluted step 1:2. Flask expressed nAbs were tested at a starting concentration of 2 µg ml$^{-1}$ and diluted step 1:2. Technical duplicates and triplicates were performed to evaluate the IC$_{100}$ of TAP and purified nAbs, respectively. In each plate positive and negative control were used as previously described[16,17,21,22].

### SARS-CoV-2 virus variants CPE-MN neutralization assay

The SARS-CoV-2 viruses used to perform the CPE-MN neutralization assay were the Wuhan (SARS-CoV-2/INMI1-Isolate/2020/Italy: MT066156), Omicron BA.1 (GISAID ID: EPI_ISL_6794907), BA.2 (GISAID ID: EPI_ISL_10654979), BA.4 (GISAID ID: EPI_ISL_13360709) and BA.5 (GISAID ID: EPI_ISL_13389618).

### Single cell RT-PCR and Ig gene amplification and transcriptionally active PCR expression

Previously obtained PCRII products[16,17] were used to recover the antibody heavy and light chain sequences, through Sanger sequencing, and for antibody transcriptionally active PCR (TAP) expression into recombinant IgG1[37]. TAP reaction was performed using 5 µL of Q5 polymerase (NEB), 5 µL of GC Enhancer (NEB), 5 µL of 5X buffer, 10 mM dNTPs, 0.125 µL of forward/reverse primers and 3 µL of ligation product, using the following cycles: 98°/2′, 35 cycles 98°/10″, 61°/20″, 72°/1′ and 72°/5′. TAP products were purified and subsequently quantified by Qubit Fluorometric Quantitation assay (Invitrogen). Transient transfection was performed using Expi293F cell line (Thermo Fisher) following manufacturing instructions.

### Flask expression and purification of human monoclonal antibodies

Plasmids carrying the antibody heavy and light chain of nAbs were used for transient transfection of Expi293F™ cells (Thermo Fisher) as previously described[22]. Briefly, cells were grown for 6 days at 37 °C with 8% CO$_2$ shaking at 125 rpm according to the manufacturer's protocol. Six days after transfection, cell cultures were harvested and clarified by centrifugation (248 × $g$, for 8 min at RT). Cell supernatants were recovered, filtered with 0.45 µm filters to remove particulate material, and then purified through affinity chromatography using a 1 mL HiTrap Protein G HP column (GE Healthcare Life Sciences). Antibodies were eluted from the column using 0.1 M glycine-HCl, pH 2.7. Protein-containing fractions were pooled and dialyzed in PBS buffer overnight at 4 °C. Final antibody concentrations were determined by measuring the A562 using Pierce™ BCA Protein Assay Kit (Thermo Scientific). Purified antibodies were stored at −80 °C prior to use.

### Functional repertoire analyses

nAbs VH and VL sequence reads were manually curated and retrieved using CLC sequence viewer (Qiagen). Aberrant sequences were removed from the data set. Analyzed reads were saved in FASTA format and the repertoire analyses was performed using Cloanalyst (http://www.bu.edu/computationalimmunology/research/software/)[38,39].

### Alluvial plot of germline neutralization potency distribution

Alluvial plots were generated to display the neutralization potency distribution of IGHV;IGHJ germlines among the three analyzed cohorts: seronegative 2nd dose (SN2), 3rd dose (SN3), and

seropositive 2nd dose (SP2). For each variant indicated on the ordinates, six different categories (strata) are represented to group antibody neutralization potency depending on the average germline $IC_{100}$: extremely neutralizing ($\leq$10 ng ml$^{-1}$), high neutralizing ($\leq$100 ng ml$^{-1}$), medium neutralizing ($\leq$1000 ng ml$^{-1}$), low neutralizing ($\leq$10,000 ng ml$^{-1}$), very low neutralizing ($<$100,000 ng ml$^{-1}$) and not neutralizing ($\geq$100,000 ng ml$^{-1}$). The germline frequency for each single strata for each variant is represented by the flow size. For the functional antibody repertoire analyses of these two groups, we highlighted the V-J gene rearrangements of nAbs representing at least 10% of all antibodies able to cross-neutralize a specific Omicron variant. Below this threshold, several germlines showed identical or similar frequency values and therefore, were not considered as predominant. Selected germlines were colored in light blue, dark blue, and red for SN2, SN3, and SP2, respectively. The figure was assembled with ggplot2 v3.3.5.

## Statistical analysis

Statistical analysis was assessed with GraphPad Prism Version 8.0.2 (GraphPad Software, Inc., San Diego, CA). Nonparametric Mann-Whitney t test was used to evaluate statistical significance between the two groups analyzed in this study. Statistical significance was shown as * for values $\leq$ 0.05, ** for values $\leq$ 0.01, and *** for values $\leq$ 0.001.

## Reporting summary

Further information on research design is available in the Nature Portfolio Reporting Summary linked to this article.

## Data availability

Source data are provided with this paper. All data supporting the findings in this study are available within the article or can be obtained from the corresponding author upon request. SARS-CoV-2 antibody sequences were deposited and accessible from https://github.com/dasch-lab/SARS-CoV-2_nAb_third_dose.

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

## Acknowledgements

This work received funding by the European Research Council (ERC) advanced grant (agreement number 787552 (vAMRes)), and the Italian Ministry of Health (COVID-2020-12371817 project). In addition, this work was supported by a fundraising activity promoted by Unicoop Firenze, Coop Alleanza 3.0, Unicoop Tirreno, Coop Centro Italia, Coop Reno e Coop Amiatina. Piet Maes acknowledges support from the Research Foundation Flanders (COVID19 research grant G0H4420N) and Internal Funds KU Leuven (grant 3M170314).

## Author contributions

Conceived the study: E.A. and R.R.; TAP and flask expression of monoclonal antibodies: I.P., G.A., and V.A.; Recovered VH and VL sequences and performed the repertoire analyses: P.P., E.A. and G.M.; Performed neutralization assays in BSL3 facilities: E.A., I.P., G.Pie., G.A., G.Pic., L.B., and G.G.; Supported day-by-day laboratory activities and management: C.D.S.; Manuscript writing: E.A. and R.R.; Final revision of the manuscript: E.A., I.P., G.Pie., G.M., G.A., V.A., P.P., L.B., G.G., G.Pic., C.D.S., C.S., D.M., E.M., P.M., and R.R.; Coordinated the project: E.A., D.M., E.M., P.M., and R.R.

## Competing interests

E.A., I.P., V.A., P.P., C.D.S., C.S., and R.R. are listed as inventors of full-length human monoclonal antibodies described in Italian patent applications n. 102020000015754 filed on June 30th 2020, 102020000018955 filed on August 3rd 2020 and 102020000029969 filed on 4th of December 2020, and the international patent system number PCT/IB2021/055755 filed on the 28th of June 2021. All patents were submitted by Fondazione Toscana Life Sciences, Siena, Italy. Remaining authors have no competing interests to declare.
