## [Peer Review File · Nature Communications]

mRNA vaccines and hybrid immunity use different B cell germlines against Omicron BA.4 and BA.5REVIEWER COMMENTS

Reviewer #1 (Remarks to the Author):

In this paper Andreano and colleagues examine the neutralization potential against BA.4 and BA.5 of 2 panels of human monoclonal antibodies isolated from either seronegative individuals given 2-3 doses of mRNA vaccine (SN2-SN3) or individuals that were vaccinated following a natural SARS-CoV-2 infection (SP2).

None of the mAb isolated from SN2 individuals neutralized BA.4/5 whilst around 15% from the SN3 and SP2 could neutralize BA.4/5.

The authors go on to map the antigenic sites bound by the mAb. For the SN2 and SN3 groups antibodies bind mainly to the NTD and RBD class 1/2 epitope, whilst interestingly BA.4/5 neutralizing mAb from the SP2 group preferentially targeted the class 3 epitope and NTD. Sequencing of heavy chain variable regions further exemplified the difference between the SN2/3 and DP2 groups which showed major differences in germline usage.

Overall this is a simple paper that shows that hybrid immunity imparted by natural and vaccine challenge differs from simple 2 and 3 dose vaccination, producing different B cell repertoires for unknown reason. Of note I found Fig. 3 quite hard to follow.

Reviewer #2 (Remarks to the Author):

In their manuscript Andreano and colleagues describe differences in the monoclonal antibody response to SARS-CoV-2 Omicron variants depending on exposure history. This is an interesting and wonderful paper. However, there are some points that need the authors' attention.

Major points

1) Several studies have observed increased protection against Omicron in individuals with hybrid immunity compared to naïve individuals with three vaccine shots. Furthermore, a recent study in Qatar has shown that a single infection may still provide significant protection against Omicron reinfection – including BA.5 – while vaccination does not. Now, it would be good to incorporate this into the discussion.

2) In addition, it would be good to specify if any of the mAbs isolated from the cohorts were originally IgA. That would be helpful to understand further differences between the cohorts.

Minor points

1) Typically, all variants start with a capital letter (e.g. Omicron).

2) Abbreviations are partially not defined in the abstract, the main article and the methods. Please also define SARS-CoV-2 and COVID-19.

3) Line 39: 'spike', not 'Spike'

- 4) Line 74: Remove 'a'.
- 5) Line 127: Remove ','.
- 6) Line 180: 'was', not 'were'
- 7) Line 198: 'drives', not 'drive'
- 8) Line 205: Should be 'differences in its'.
- 9) Line 208: The antibody response cannot be induced by immunity, only by vaccination or infection. Please correct this.
- 10) Line 287: There are no 'COVID-19 infections', there can only be 'SARS-CoV-2 infections'.
- 11) Line 294: This sentence sounds weird. Consider reformulating it.
- 12) Line 307: 'tissue culture infectious dose', not 'Tissue Culture Infectious Dose'.
- 13) Line 334: It is OK to specify rpm, but then the rotor size needs to be given too. Otherwise specify rcf.

Reviewer #3 (Remarks to the Author):

In this study, the author tested the neutralizing activity of 482 antibodies against different Omicron sublineages (BA.1, BA.2, BA.4, and BA.5). These antibodies were isolated from donors with 2 or 3 mRNA vaccine doses (SN2 and SN3, respectively), as well as people that had been vaccinated after infection (SP2). The results showed that antibodies from SN3 and SP2 are much better than those from SN2 in terms of neutralizing Omicron sublineages. Additional analyses showed that antibodies from SN2/3 and SP2 have different preferences in epitopes and germline gene usages. Overall, this study is straightforward and data are well-presented. The results contribute to the understanding of SARS-CoV-2 antibody immunity. However, there are several concerns that need to be addressed.

Major comments

1. In Fig. 1, the calculation of geometric mean 100% inhibitory concentration (GM-IC100) did not include antibodies that were "not-neutralizing", which is a bit misleading and creates some confusions. For example:

- In panel A, the differences between Wuhan and BA.1 as well as between Wuhan and BA.2 are reported to be not significant. However, based on the distribution of the data points, there are dramatic differences (e.g. 0% not neutralizing for Wuhan vs 98.1% not neutralizing for BA.1).
- In panel D, the GM-IC100 of SN2 to BA.2 is lower (i.e. more potent) than SN3/SP2 to BA.2. Similarly, SN2 is more potent against BA.2 than Wuhan according to GM-IC100.

2. Lines 95-96, "The SN2 group (n = 46) showed ... A similar trend was also observed for RBD and NTD-targeting nAbs isolated from the SN3 cohort (n = 197)." According to the

donor ID in Fig. 3a-b, it seems like all the donors in the SN3 cohort (VAC-001, VAC-002, VAC-008, and VAC-010) also belong to the SN2 cohort? If that is the case, the finding stated in lines 95-96 is expected.

3. Lines 202-204, "The observation that homologous mRNA vaccination and infection drive the expansion of different B cell germlines which produce nAbs targeting distinct epitopes on the SARS-CoV-2 S protein ..." I am not sure how valid this statement is, given that there seems to be a high heterogeneity of germline gene usage among donors within each cohort and there are only 4 to 5 donors are in each cohort (Fig. 3a-c).

Minor comments

1. Line 78, "Fig. 1b-c" should only be "Fig. 1b"
2. Lines 80-81, Fig. 1c should be cited.

Reviewer #4 (Remarks to the Author):

Summary and main findings

Given the extensive immune escape of the Omicron variants, it is vital to understand how antibodies retain neutralising activity against Omicron if generated by hybrid immunity or boosted by an additional vaccine. In this manuscript, the authors compare antibodies in three populations: those who have been received 2 or 3 vaccine doses but remain uninfected, or those with hybrid immunity. Interestingly, the results here clearly show that three doses of an mRNA vaccine elicit a different antibody/B cell response compared to hybrid immunity, with respect to both epitope regions targeted and B cell germlines involved. This manuscript is exciting, but needs to address some significant and minor points:

Major comments

Could the authors please comment on how many antibodies came from each participant? Did nAbs from the same individual have similar properties? e.g. Were the Class 3 nAbs produced by all participants in the SP2 group or only one or two individuals?

Was the previous classification of RBD-targeting nAbs done on this same panel of 482 antibodies (lines 90-93)?

The numbers of antibodies described in Figure 2 (Lines 93-97) do not correspond to the numbers for Figure 1. Can the authors please explicitly state which antibodies from Figure one were included in Figure 2 as nAbs targeting the RBD or NTD regions?

The "impact on therapeutics" section is valuable to the manuscript and should be added to the main figures. Therapeutics would also be interesting to expand upon in the discussion. Given the findings described in this manuscript- where do we go from here? What are considerations for new therapeutic antibodies or vaccines? Can the authors also speculate on how omicron-specific vaccines might fit into this?

Since the timing of SARS-CoV-2 waves and variants differed between regions, what variant was circulating when the SP2 group was infected?

Can the authors please include some clinical data on the participants which may affect immunity e.g. co-morbidities, age and sex? Details on disease severity and pathology (e.g. long covid) on the SP2 group should also be included if available.

How confident are the authors that the other participants (SN2 or SN3) did not have an asymptomatic infection given the fast waning of N antibodies?

A significant limitation of the study is the low number (and diversity) of participants, which should be discussed.

Can the authors please briefly describe the method to generate PCRII products used to obtain antibody heavy and light chain sequences? The “single cell” level of this analysis is highlighted throughout the manuscript, thus additional detail is required.

Minor comments

Lines 82-85: This section of reported results is confusing, and it isn't clear which fold change corresponds to which variant or which participant group.

There are some grammatical or typographical errors which make the text difficult to understand e.g. lines 137-140- are the authors referring to neutralisation potency or abundance? Line 51-52 “As observed with all previous.....”

Which virus is the sentence on line 140-141 referring to?

Line 301- Do the authors mean “live” virus neutralisation assay?

The methods section line 359-360 refers to a “heatmap on the left” which is likely referring to Figure 3 and should be included in the figure legend.

REBUTTAL: NCOMMS-22-31562 - mRNA vaccines and hybrid immunity use different B cell germlines to neutralize Omicron BA.4 and BA.5

General comment

We thank the editor and reviewers for their critical assessment of our work and for the positive feedbacks herein provided. In the revised version of the manuscript, we amended the text and figures in accordance with the reviewers' comments. In addition, you can find below a point-by-point response to all comments.

REVIEWER COMMENTS

Reviewer #1 (Remarks to the Author):

In this paper Andreano and colleagues examine the neutralization potential against BA.4 and BA.5 of 2 panels of human monoclonal antibodies isolated from either seronegative individuals given 2-3 doses of mRNA vaccine (SN2-SN3) or individuals that were vaccinated following a natural SARS-CoV-2 infection (SP2). None of the mAb isolated from SN2 individuals neutralized BA.4/5 whilst around 15% from the SN3 and SP2 could neutralize BA.4/5. The authors go on to map the antigenic sites bound by the mAb. For the SN2 and SN3 groups antibodies bind mainly to the NTD and RBD class 1/2 epitope, whilst interestingly BA.4/5 neutralizing mAb from the SP2 group preferentially targeted the class 3 epitope and NTD. Sequencing of heavy chain variable regions further exemplified the difference between the SN2/3 and SP2 groups which showed major differences in germline usage. Overall this is a simple paper that shows that hybrid immunity imparted by natural and vaccine challenge differs from simple 2 and 3 dose vaccination, producing different B cell repertoires for unknown reason.

R1: Of note I found Fig. 3 quite hard to follow.

A1: The authors understand the complexity of figure 3 but we believe that this representation better summarizes the complex biological scenario of B cell germline usage in the immunologically distinct cohorts analyzed in this study. To support the data shown in Fig. 3, we generated supplementary table 2 and 3 and described these data in detail in the result section. In addition, to facilitate the interpretation of this figure, we have added germlines details in the figure 3 caption (line 460-463 of the revised version of this manuscript).

Reviewer #2 (Remarks to the Author):

In their manuscript Andreano and colleagues describe differences in the monoclonal antibody response to SARS-CoV-2 Omicron variants depending on exposure history. This is an interesting and wonderful paper. However, there are some points that need the authors' attention.

Major points

R1: Several studies have observed increased protection against Omicron in individuals with hybrid immunity compared to naïve individuals with three vaccine shots. Furthermore, a recent study in Qatar has shown that a single infection may still provide significant protection against Omicron reinfection – including BA.5 – while vaccination does not. Now, it would be good to incorporate this into the discussion.

A1: We thank the reviewer for this comment. We extended our discussion section (line 194-201), to cover this point. In addition, the Qatar study highlighted by this reviewer (DOI: 10.1056/NEJMc2209306) has been incorporated in our references (Ref. 34). We believe that the main difference that can explain differences of our work with other studies is the depth of our analysis. Indeed, in our study we interrogated exclusively the repertoire of neutralizing antibodies while previous works evaluated the polyclonal antibody response or percentage of reinfection in subjects vaccinated or with hybrid immunity.

R2: In addition, it would be good to specify if any of the mAbs isolated from the cohorts were originally IgA. That would be helpful to understand further differences between the cohorts.

A2: We did not make any comparison on IgAs from the different cohorts as only 4/58 (6.9%), 27/278 (9.7%) and 5/288 (1.7%) IgA sequences were recovered from SN2, SP2 and SN3 respectively. In addition, we did not manage to recover and express all IgA, therefore neutralization data are not available for these antibodies. The low number of IgAs recovered does not allow to further understand the differences between SN3 and SP2.

Minor points

R3: Typically, all variants start with a capital letter (e.g. Omicron).

A3: All variants name now start with the capital letter.

R4: Abbreviations are partially not defined in the abstract, the main article and the methods. Please also define SARS-CoV-2 and COVID-19.

A4: The abbreviations highlighted by this reviewer have now been defined.

R5: Line 39: 'spike', not 'Spike'

A5: The text was amended accordingly.

R6: Line 74: Remove 'a'.

A6: The text was amended accordingly.

R7: Line 127: Remove ','.

A7: The text was amended accordingly.

R8: Line 180: 'was', not 'were'

A8: The text was amended accordingly.

R9: Line 198: 'drives', not 'drive'

A9: The text was amended accordingly.

R10: Line 205: Should be 'differences in its'.

A10: The text was amended accordingly.

R11: Line 208: The antibody response cannot be induced by immunity, only by vaccination or infection. Please correct this.

A11: The text was amended accordingly.

R12: Line 287: There are no 'COVID-19 infections', there can only be 'SARS-CoV-2 infections'.

A12: The text was amended accordingly.

R13: Line 294: This sentence sounds weird. Consider reformulating it.

A13: The sentence has been reformulated.

R14: Line 307: 'tissue culture infectious dose', not 'Tissue Culture Infectious Dose'.

A14: The text was amended accordingly.

R15: Line 334: It is OK to specify rpm, but then the rotor size needs to be given too. Otherwise specify rcf.

A15: Rotor and its size were provided in the M&M section (line 371).

Reviewer #3 (Remarks to the Author):

In this study, the author tested the neutralizing activity of 482 antibodies against different Omicron sublineages (BA.1, BA.2, BA.4, and BA.5). These antibodies were isolated from donors with 2 or 3 mRNA vaccine doses (SN2 and SN3, respectively), as well as people that had been vaccinated after infection (SP2). The results showed that antibodies from SN3 and SP2 are much better than those from SN2 in terms of neutralizing Omicron sublineages. Additional analyses showed that antibodies from SN2/3 and SP2 have different preferences in epitopes and germline gene usages. Overall, this study is straightforward and data are well-presented. The results contribute to the understanding of SARS-CoV-2 antibody immunity. However, there are several concerns that need to be addressed.

Major comments

R1: In Fig. 1, the calculation of geometric mean 100% inhibitory concentration (GM-IC₁₀₀) did not include antibodies that were “not-neutralizing”, which is a bit misleading and creates some confusions. For example:

- In panel A, the differences between Wuhan and BA.1 as well as between Wuhan and BA.2 are reported to be not significant. However, based on the distribution of the data points, there are dramatic differences (e.g. 0% not neutralizing for Wuhan vs 98.1% not neutralizing for BA.1).
- In panel D, the GM-IC₁₀₀ of SN2 to BA.2 is lower (i.e. more potent) than SN3/SP2 to BA.2. Similarly, SN2 is more potent against BA.2 than Wuhan according to GM-IC₁₀₀.

A1: We thank the reviewer for these comments. Figure 1 aimed to evaluate the different neutralization potency among neutralizing antibodies, therefore, antibodies that did not retained neutralization activity were not included in the analyses. This has been now highlighted in the result section (line 86-87). We could have not included not-neutralizing mAbs in the analyses as they do not have an IC₁₀₀ value that can be used. A fictitious value for all not-neutralizing antibodies would have led to a more misleading analysis. As for panel A, lack of significance is mainly due by the differences among size groups. For Wuhan vs BA.1, the Mann-Whitney test requires at least two values in each group to be performed, therefore no significance could be evaluated. While for Wuhan vs BA.2 no significance was found (p=0.1529). For panel D, it is true that the GM-IC₁₀₀ of SN2 is the highest among tested cohorts, but we should consider that only 4/52 (7.7%) nAbs retained their neutralization activity. Low numbers of nAbs against a specific SARS-CoV-2 variant reduces the reliability of the overall GM-IC₁₀₀ evaluated in the group. Indeed, it is well described that the polyclonal antibody response in subjects immunized with two vaccine doses are the least functional when compared to three vaccine doses or hybrid immunity (<https://doi.org/10.1016/j.chom.2022.07.002>, Figure 2; <https://doi.org/10.1056/NEJMc2206576>, Figure 1). The overall neutralization titers are also dictated by the abundance of neutralizing antibodies in the polyclonal response and not only by the potency of a small fraction of nAbs that still retain neutralization activity.

R2: Lines 95-96, “The SN2 group (n = 46) showed ... A similar trend was also observed for RBD and NTD-targeting nAbs isolated from the SN3 cohort (n = 197).” According to the donor ID in Fig. 3a-b, it seems like all the donors in the SN3 cohort (VAC-001, VAC-002, VAC-008, and VAC-010) also belong to the SN2 cohort? If that is the case, the finding stated in lines 95-96 is expected.

A2: The four donors in the SN3 cohort are the same of the SN2 group. We now have highlighted this point in both the results (line 71-72) and M&M (line 332-333) sections. In addition, “as expected” was also included in the text (line 104).

R3: Lines 202-204, “The observation that homologous mRNA vaccination and infection drive the expansion of different B cell germlines which produce nAbs targeting distinct epitopes on the SARS-CoV-2 S protein ...”

I am not sure how valid this statement is, given that there seems to be a high heterogeneity of germline gene usage among donors within each cohort and there are only 4 to 5 donors are in each cohort (Fig. 3a-c).

A3: We agree with this reviewer that a broader sample size in the three different cohort analyzed would have given a higher power to this study. Despite that, the depth of the analyses performed in our work is extremely difficult to expand to numerous donors given the complexity of the workflow and the time needed to perform the experiments and analyze the data. The time needed to expand our cohorts does not fit with the pace that needs to be maintained to quickly evaluate the antibody response against emerging variants. To mitigate this problem, for the comparison of B cell germline usage between SN3 and SP2 (the two major groups that we described in our study) we highlighted only those germlines that were found in multiple donors within each cohort. In addition, we have stated in the discussion section that a limit of this study is the is the low number of participants (line 201-203).

Minor comments

R4: Line 78, “Fig. 1b-c” should only be “Fig. 1b”

A4: The text was amended accordingly.

R5: Lines 80-81, Fig. 1c should be cited.

A5: The text was amended accordingly.

Reviewer #4 (Remarks to the Author):

Summary and main findings

Given the extensive immune escape of the Omicron variants, is vital to understand how antibodies retain neutralising activity against Omicron if generated by hybrid immunity or boosted by an additional vaccine. In this manuscript, the authors compare antibodies in three populations: those who have been received 2 or 3 vaccine doses but remain uninfected, or those with hybrid immunity. Interestingly, the results here clearly show that three doses of an mRNA vaccine elicit a different antibody/B cell response compared to hybrid immunity, with respect to both epitope regions targeted and B cell germlines involved. This manuscript is exciting, but needs to address some significant and minor points:

Major comments

R1: Could the authors please comment on how many antibodies came from each participant? Did nAbs from the same individual have similar properties? e.g. Were the Class 3 nAbs produced by all participants in the SP2 group or only one or two individuals?

A1: We thank the reviewer for this comment. Each individual shows heterogeneity within their respective antibody response, but similar trends of classes distribution were observed among subjects within the same cohort. This can be observed from our previous works (reference 16 and 17). In particular, for SN2 and SP2, this was described in the Figure 1a-b, Extended Data Fig. 7a and Extended Data Table 3 (<https://doi.org/10.1038/s41586-021-04117-7>; Ref 16), while for SN3 the antibody distribution per each individual was described in Figure 2a and Extended Data Fig. 2 (<https://doi.org/10.1101/2022.05.09.491201>; Ref 17). We highlighted this point in the results section line 99-101.

R2: Was the previous classification of RBD-targeting nAbs done on this same panel of 482 antibodies (lines 90-93)?

A2: Yes, the same classification was used for all nAbs previously tested. We have added this statement at line 97. In addition, in references 16 and 17 it is possible to observe the nAb classification strategy used for all cohorts. Specifically, the classification for SN2 and SP2 can be observed in Ref. 16, Extended Data Fig. 6 and Extended Data Fig. 7a (<https://doi.org/10.1038/s41586-021-04117-7>), while the same classification for SN3 can be found in Ref. 17 Figure 2a (<https://doi.org/10.1101/2022.05.09.491201>).

R3: The numbers of antibodies described in Figure 2 (Lines 93-97) do not correspond to the numbers for Figure 1. Can the authors please explicitly state which antibodies from Figure one were included in Figure 2 as nAbs targeting the RBD or NTD regions?

A3: As highlighted in the paragraph title “Mapping RBD and NTD cross-protective nAbs” and at line 94-95 of the revised version of this manuscript, the mapping analyses of nAbs was performed exclusively on RBD and NTD targeting antibodies. Therefore, nAbs that did not target these two domains were not included in the analysis. As suggested by this reviewer, we further stressed this point in the revised version of this manuscript (line 95-96).

R4: The “impact on therapeutics” section is valuable to the manuscript and should be added to the main figures. Therapeutics would also be interesting to expand upon in the discussion. Given the findings described in this manuscript- where do we go from here? What are considerations for new therapeutic antibodies or vaccines? Can the authors also speculate on how omicron-specific vaccines might fit into this?

A4: As suggested by this reviewer, previous “Supplementary Figure 3” has now been added in the main text as “Figure 4”. In addition, we expanded the discussion section to cover the questions highlighted in this comment (line 221-233).

R5: Since the timing of SARS-CoV-2 waves and variants differed between regions, what variant was circulating when the SP2 group was infected?

A5: Subjects in the SP2 cohort resulted positive to SARS-CoV-2 infection between October and November 2020 when in Italy only the D614G SARS-CoV-2 variant was circulating. This statement, and related references, have been added in the M&M section line 325-327 of the revised manuscript.

R6: Can the authors please include some clinical data on the participants which may affect immunity e.g. co-morbidities, age and sex? Details on disease severity and pathology (e.g. long covid) on the SP2 group should also be included if available.

A6: Information on sex, age and disease severity have been added to the M&M paragraph “Enrollment of COVID-19 vaccinees and human sample collection” line 317-366 of the revised version of this manuscript.

R7: How confident are the authors that the other participants (SN2 or SN3) did not have an asymptomatic infection given the fast waning of N antibodies?

A7: Infection was excluded from SN3 as no SARS-CoV-2 infection was reported. Anti-N screening was not performed, but the neutralizing antibodies repertoire analyses reported in the Extended Data Figure 6 of our previous work (<https://doi.org/10.1101/2022.05.09.491201>), showed extremely similar profiles among subjects, further suggesting that subjects were not exposed to SARS-CoV-2 infection.

R8: A significant limitation of the study is the low number (and diversity) of participants, which should be discussed.

A8: As suggested by this reviewer, we included this limitation of the study in the discussion section (line 201-203)

R9: Can the authors please briefly describe the method to generate PCRII products used to obtain antibody heavy and light chain sequences? The “single cell” level of this analysis is highlighted throughout the manuscript, thus additional detail is required.

A9: The nAbs evaluated in this work derive from our previous studies which are referenced throughout the text (Ref. 16, 17 and 22). Monoclonal antibody sequences were not retrieved in this work, therefore we could

not report these methods within this manuscript. Anyway, the whole procedure has been described in detail in references 16 and 17 provided in the M&M section at line 359.

Minor comments

R10: Lines 82-85: This section of reported results is confusing, and it isn't clear which fold change corresponds to which variant or which participant group.

A10: This part of the text has been reworded (line 85-91).

R11: There are some grammatical or typographical errors which make the text difficult to understand e.g. lines 137-140- are the authors referring to neutralisation potency or abundance? Line 51-52 "As observed with all previous....."

A11: The text was modified to improve to improve its readability.

R12: Which virus it the sentence on line 140-141 referring to?

A12: It referred to all Omicron variants tested in this study. The text was modified to improve its readability.

R13: Line 301- Do the authors mean "live" virus neutralisation assay?

A13: "authentic" was replaced with "live" virus.

R14: The methods section line 359-360 refers to a "heatmap on the left" which is likely referring to Figure 3 and should be included in the figure legend.

A14: The sentence highlighted in this comment was moved in the Figure 3 legend (line 456-457).

REVIEWERS' COMMENTS

Reviewer #2 (Remarks to the Author):

The authors have addressed the reviewers' comments well.

Reviewer #3 (Remarks to the Author):

The authors have addressed all my previous concerns.

Reviewer #4 (Remarks to the Author):

The authors compare antibodies in three populations: those who have been received 2 or 3 vaccine doses but remain uninfected, or those with hybrid immunity. They report that the epitope regions targeted by the antibodies elicited by 3 vaccine doses suffered from that elicited by hybrid immunity. This is particularly relevant given recent reports that hybrid immunity is more effective compared to vaccine-induced responses

The authors have addressed all queries sufficiently and added the requested clarifications and information to the manuscript.

Point-by-point response: NCOMMS-22-31562A - mRNA vaccines and hybrid immunity use different B cell germlines against Omicron BA.4 and BA.5

General comment

We thank the editor and reviewers for their critical assessment of our work and for the positive feedbacks herein provided. In the revised version of the manuscript, we amended the text and figures in accordance with the journal policies and editors' comments. In addition, you can find below a point-by-point response to the referees' comments.

REVIEWER COMMENTS

Reviewer #2 (Remarks to the Author):

R1: The authors have addressed the reviewers' comments well.

A1: We thank this reviewer for the assessment of our work and for this final comment.

Reviewer #3 (Remarks to the Author):

R1: The authors have addressed all my previous concerns.

A1: We thank this reviewer for the assessment of our work and for this final comment.

Reviewer #4 (Remarks to the Author):

R1: The authors compare antibodies in three populations: those who have been received 2 or 3 vaccine doses but remain uninfected, or those with hybrid immunity. They report that the epitope regions targeted by the antibodies elicited by 3 vaccine doses suffered from that elicited by hybrid immunity. This is particularly relevant given recent reports that hybrid immunity is more effective compared to vaccine-induced responses. The authors have addressed all queries sufficiently and added the requested clarifications and information to the manuscript.

A1: We thank this reviewer for the assessment of our work and for this final comment.